# DNA metabarcoding and spatial modelling link diet diversification with distribution homogeneity in European bats

Antton Alberdi[1✉], Orly Razgour [2,3], Ostaizka Aizpurua[1], Roberto Novella-Fernandez[2], Joxerra Aihartza [4], Ivana Budinski [5], Inazio Garin[4], Carlos Ibáñez [6], Eñaut Izagirre [4,7], Hugo Rebelo [8,9], Danilo Russo [10], Anton Vlaschenko [11], Violeta Zhelyazkova[12], Vida Zrnčić[13] & M. Thomas P. Gilbert[1,14]

Inferences of the interactions between species' ecological niches and spatial distribution have been historically based on simple metrics such as low-resolution dietary breadth and range size, which might have impeded the identification of meaningful links between niche features and spatial patterns. We analysed the relationship between dietary niche breadth and spatial distribution features of European bats, by combining continent-wide DNA metabarcoding of faecal samples with species distribution modelling. Our results show that while range size is not correlated with dietary features of bats, the homogeneity of the spatial distribution of species exhibits a strong correlation with dietary breadth. We also found that dietary breadth is correlated with bats' hunting flexibility. However, these two patterns only stand when the phylogenetic relations between prey are accounted for when measuring dietary breadth. Our results suggest that the capacity to exploit different prey types enables species to thrive in more distinct environments and therefore exhibit more homogeneous distributions within their ranges.

[1] Centre for Evolutionary Hologenomics, GLOBE Institute, Faculty of Health and Medical Sciences, University of Copenhagen, 1353 Copenhagen, Denmark. [2] Biological Sciences, University of Southampton, Southampton SO17 1BJ, UK. [3] Biosciences, University of Exeter, Exeter EX4 4PY, UK. [4] University of the Basque Country UPV/EHU, 48940 Leioa, Spain. [5] Department of Genetic Research, Institute for Biological Research "Siniša Stanković", University of Belgrade, Belgrade 11060, Serbia. [6] Doñana Biological Station, CSIC, 41092 Seville, Spain. [7] Basque Centre for Climate Change BC3, 48940 Leioa, Spain. [8] CIBIO-InBIO, Centro de Investigação em Biodiversidade Recursos e Genéticos, Universidade do Porto, Vairão, Portugal. [9] CIBIO-InBIO, Instituto Superior de Agronomia, Universidade de Lisboa, 1349-017 Lisboa, Portugal. [10] Wildlife Research Unit, Dipartimento di Agraria, Università degli Studi di Napoli Federico II, 80055 Portici, Naples, Italy. [11] Bat Rehabilitation Center of Feldman Ecopark, Kharkiv 62340, Ukraine. [12] National Museum of Natural History, Bulgarian Academy of Sciences, 1 Tsar Osvoboditel Blvd., 1000 Sofia, Bulgaria. [13] Croatian Biospeleological Society, HR-10000 Zagreb, Croatia. [14] Norwegian University of Science and Technology, University Museum, 7491 Trondheim, Norway. ✉email: antton.alberdi@sund.ku.dk

The study of the relationship between the ecological niche breadth and spatial distribution of species has been a core topic in ecology, since Hutchinson's initial conceptualisation of the ecological niche[1–3]. Ecological niche breadth measures the degree of specialisation or generalisation of the resources species use (Eltonian definition of ecological niche) or the conditions which they inhabit (Grinnellian definition)[4,5]. Different domains of the ecological niche, such as climatic tolerance, habitat breadth and dietary breadth, have been shown to be positively associated with species' geographical range sizes[3], which is the feature historically employed to characterise the spatial distribution of species[2]. The relationship between ecological niche breadth and range size is based on the idea that species that use a greater array of resources, and thrive under a wider variety of conditions, should become more widespread[2,6]. However, the effect size of the correlations gradually decreases when transitioning from Grinnellian (e.g. climatic tolerance) to Eltonian (e.g. diet) niche features[3]. In fact, a meta-analysis of 20 independent studies showed that dietary niche breadth is only positively related to range size in arthropods, but not in vertebrates[3].

The lack of a consistent link between dietary niche breadth and range size in vertebrates could be due to their broader dietary niche breadth in comparison to that of the more thoroughly studied phytophagous arthropods, which typically exhibit a larger degree of resource specialisation[7,8]. In addition, vertebrates often exhibit mismatches between different niche components; they can be generalists for dietary resources, but specialists for some other resources, such as roosts[9,10]. These discrepancies could lead to discordance between the given Eltonian features and range size. Finally, range size is also known to be largely dependent on the historic legacy of speciation, Quaternary climatic changes, and the dispersal capacity of species[11].

Besides range size, other spatial features might also be related to dietary niche breadth. The homogeneity of the spatial distribution, i.e. how evenly animals are distributed within their geographic range[12], is one of them. It has been hypothesised that a generalist species (e.g. wide dietary niche) will be able to find useful resources in more different environments than a specialist (e.g. narrow dietary niche), so their spatial distribution should become more homogeneous[13]. While the hypothesis that links dietary niche breadth with range size has been thoroughly studied[3], the hypothesis that relates dietary niche breadth with distribution homogeneity has received far less attention[13].

We argue that because traditionally employed metrics are excessively simplified, they might be unable to reveal ecologically meaningful relationships between dietary niche features and broad-scale spatial patterns in vertebrates. However, current analytical tools might provide the required amplitude and resolution to unravel more complex links. High-throughput DNA sequencing-based (DNA metabarcoding) and species distribution modelling (SDM) tools now enable more nuanced analysis of dietary and spatial patterns. DNA metabarcoding allows comprehensive analysis of dietary variation, by considering different components of dietary diversity, such as richness (how many prey types are consumed), evenness (the balance of the relative consumption of each prey), and regularity (the degree of similarity across consumed prey)[14,15]. SDMs predict how presence probability is distributed across the geographic range of a species, hence enabling the estimation of homogeneity of the spatial distribution.

In the present study, we contrast broad-scale dietary and spatial features of a vertebrate system, namely a European bat assemblage, with the aim of gaining further insights into the relationship between the dietary niche breadth and spatial features of vertebrates. Bats provide an excellent case study for

understanding the relationship between dietary and spatial patterns. The dietary breadth of European insectivorous bats varies considerably, from specialists on certain arthropods—usually moths—to generalists that consume a wide range of taxa[16,17]. Similarly, the spatial distribution of species also exhibits marked differences[18]. Specifically, we hypothesise that the predicted geographic distributions of species would be more homogeneous in bats with broader dietary niches.

We characterise the dietary niche of seven bat species by analysing over 400 individual faecal samples, collected at 40 locations scattered across the European continent. Faeces of each individual bat are independently analysed through DNA metabarcoding and high-throughput sequencing, using two complementary primer sets and three replicates per primer. The level of dietary specialisation is measured using the statistical framework recently developed around Hill numbers[19], by considering different components of dietary diversity: richness (dR), richness + evenness (dRE) and richness + evenness + regularity (dRER), and applying both incidence- and abundance-based approaches to quantify diet. In parallel, we generate species distribution models of the bat species, using high-quality presence records and an ensemble approach that combine different modelling algorithms. Spatial features derived from these models, such as potential range size and degree of homogeneity of the predicted suitable distributions, are also characterised based on Hill numbers applied to spatial data. The contrast between different components of dietary diversity and spatial patterns enable us to unveil relationships between dietary niche breadth and broad-scale spatial features, and test whether the hypothesis that correlates dietary niche breadth with distribution homogeneity stands in the studied European bat assemblage.

## Results

**The dietary niche of European bats is dominated by Lepidoptera and Diptera.** After applying all quality filters, the dataset included dietary information of 355 individual bats belonging to seven species (DNA sequencing details in Supplementary Table 4). Using two primer sets, we detected over 3000 different prey taxa belonging to 29 arthropod orders (Fig. 1a), though the diet of European bats was dominated by Lepidoptera and Diptera (Fig. 1b). This pattern was consistent for data generated with both primer sets (Supplementary Fig. 3), as well as both incidence- and abundance-based approaches employed to quantify dietary profiles (Supplementary Fig. 4). The reliability of the results was further confirmed through in silico taxonomic amplification bias analyses, which showed that Epp primers do not exhibit the bias towards Lepidoptera and Diptera (Supplementary Fig. 5) that is known to affect the Zeale primers[20,21]. Our results complement the existing broad-scale molecular dietary data of *Miniopterus schreibersii*[17], and provide geographically widespread molecular insights into the dietary ecology of *Myotis daubentonii*, *Myotis myotis*, *Myotis emarginatus*, *Myotis capaccinii*, *Rhinolophus euryale* and *Rhinolophus ferrumequinum*, which to our knowledge had only been studied at local scales previously[16,22–25].

**Dietary niche breadth differences depend on the components of diversity accounted for.** Dietary niche breadth measures (Fig. 1c), and species' dietary generalisation/specialisation ordination plots derived from them (Fig. 1d), were different depending on the components of diversity considered for quantifying dietary niches. Such a dependency on the considered diversity components has also been reported in other systems[26,27], and it has been attributed to the fact that each diversity component might be driven by different ecological forces[28]. For instance, the dietary niches of *M. schreibersii* and

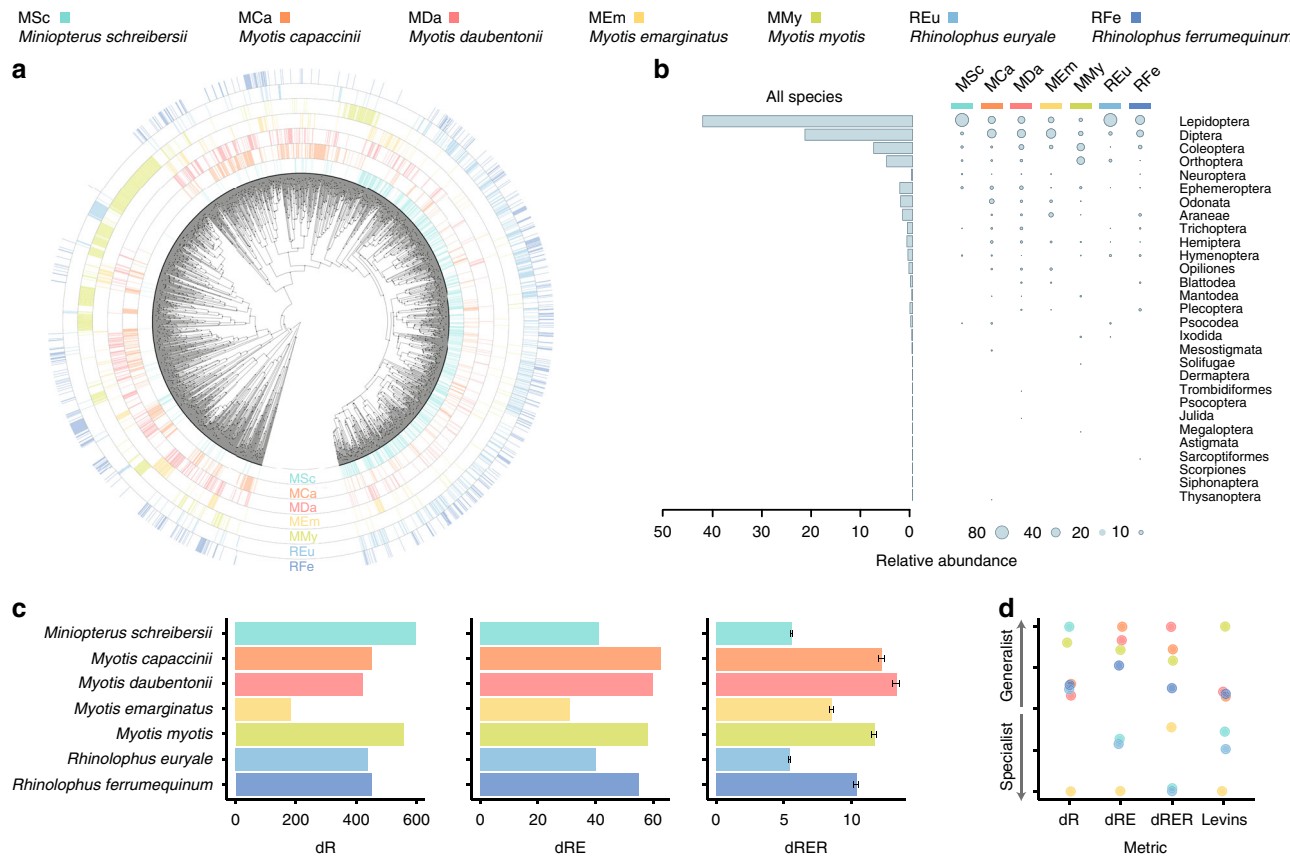

**Fig. 1 Dietary diversity statistics of the analysed bat species. a** Radial phylogenetic tree of prey detected using the Zeale primers and their occurrence patterns in each of the studied bats. A higher resolution image (Supplementary Fig. 1) and the homologous figure built from the Epp data (Supplementary Fig. 2) are available in the Supplementary Information. **b** Overall and predator species-specific representation of the arthropod taxonomic orders. The incidence-based figure is available as Supplementary Fig. 4. **c** Dietary niche breadth measures accounting for richness (dR), richness + evenness (dRE) and richness + evenness + regularity (dRER). The error bars (±SE) of dRER indicate the dispersion of the dietary niche breadths yielded when using different prey phylogenetic trees ($N = 50$) sampled from the Bayesian MCMC. **d** One-dimensional species ordination plots ranking species according to the dietary niche breadth based on different metrics. Levin's index is also included for being the most common metric employed in the literature.

*R. euryale* showed similar dRE values, yet the contribution of richness and relative evenness components differed. The dietary richness of *M. schreibersii* was almost 40% larger than *R. euryale*'s, while the evenness factor of *R. euryale* was almost 30% higher than that of *M. schreibersii*. These differences could be explained by (i) the larger home range of *M. schreibersii* compared to *R. euryale*[29,30], which might expose the former to more prey species—thus increasing dietary richness, and (ii) the higher incidence of a few locally abundant pest moth species in the diet of *M. schreibersii* than in that of *R. euryale*[17,22]—yielding lower relative evenness. As reported for other systems[31], such dietary niche differences would be overlooked if the niche breadth analyses were limited to a single diversity metric.

**Broad-scale spatial homogeneity does not correlate with range size**. Similar to dietary niche features, spatial features of species can also be measured through different metrics. To understand the relationship between the different spatial properties, we calculated the most commonly employed feature, namely range size, using IUCN cartography[32], hereafter referred to as *recognised* range size. We contrasted it with two other spatial features predicted through species' distribution modelling (Fig. 2a–g, Supplementary Table 5), namely potential range size and spatial homogeneity of the distribution. As expected, species' recognised range size within the studied region (Europe) was positively correlated with potential range size (Fig. 2h; linear model (LM):

$F = 17.36$, df = 1,5, $r^2 = 0.77$, $p$ value = 0.008). In contrast, spatial homogeneity exhibited no linear relation with range size (LM: $F = 0.91$, df = 1,5, $r^2 = 0.15$, $p$ value = 0.382). This is best illustrated by the fact that the two of the species with the highest homogeneity exhibited the largest (*M. daubentonii*) and smallest (*M. capaccinii*) recognised and potential range sizes in Europe (Fig. 2i).

**Dietary niche breadth correlates with spatial homogeneity**. The contrast between the different dietary niche breadth metrics and recognised range size was not significant (Fig. 2j, Supplementary Fig. 6). Similarly, dietary niche breadth was not correlated with spatial homogeneity when only richness and evenness components were considered (Fig. 2k, Supplementary Fig. 7; LM: $F = 0.03$, df = 1,5, $r^2 = 0.007$, $p$ value = 0.85). However, we found that the dietary niche breadth measure that accounts for all diversity components considered (dRER) was positively correlated with broad-scale spatial homogeneity of species (Fig. 2i; linear mixed model (LMM): $t = 4.95$, df = 5, $r^2_{marginal} = 0.72$, $r^2_{conditional} = 0.92$, $p$ value = 0.004). This pattern was consistent for both primer sets, the overall averaged dataset, as well as the two analytical approaches (i.e. abundance- and incidence-based) employed (Supplementary Fig. 8). According to our results, the species that consume a wider variety of distinct prey are not those that have larger range sizes, but rather those that exhibit more homogeneous spatial distribution within their ranges.

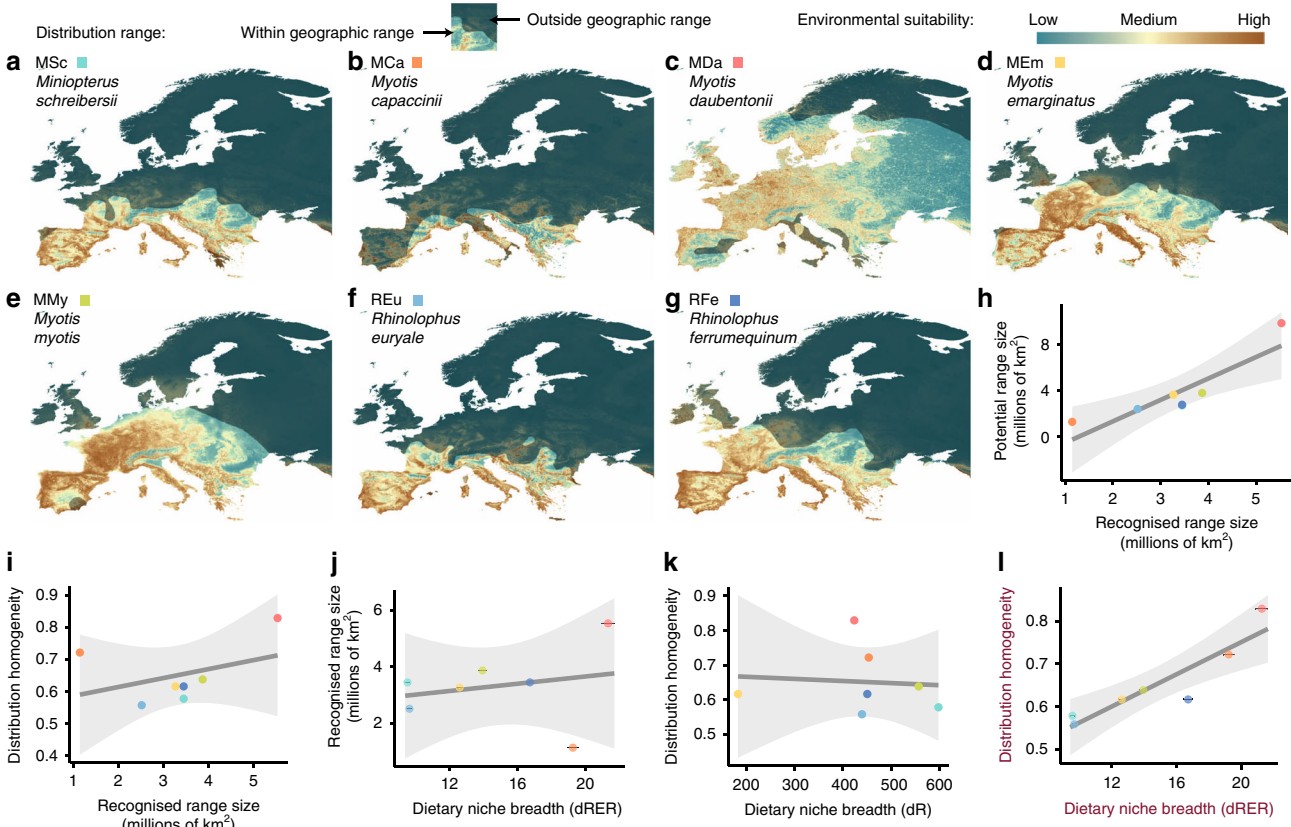

**Fig. 2 Relationship of species' dietary niche breadth measures with spatial features. a–g** Species distribution models (SDMs) of the seven bat species generated using an ensemble approach. The bright area represents the distribution area according to the IUCN Red List. **h** Significant linear relationship between the recognised range size measured from IUCN maps (x-axis) and the potential range estimated from the SDMs (y-axis). **i** Non-significant linear relationship between distribution homogeneity and distribution range. **j** Non-significant linear relationship between dietary niche breadth accounting for the three components of diversity (dRER: richness + evenness + regularity) and recognised range size. **k** Non-significant linear relationship between dietary niche breadth accounting only for dietary richness (dR) and distribution homogeneity. **l** Significant linear relationship between dietary niche breadth accounting for the three components of diversity (dRER) and distribution homogeneity. Dots indicate mean values per bat species. Note that error bars (±SE) in charts (**j**) and (**l**) indicate the dispersion of the different dietary niche breadth values yielded from the 50 iterations run with different prey phylogenetic trees to account for phylogenetic uncertainty.

**Species' dietary niche breadth is mainly driven by prey turnover across individuals.** We decomposed dietary diversity into alpha (individual bats), beta and gamma (bat species) components to gain insights into the sources of variation across species, because generalist species can be heterogeneous collections (high beta diversity) of specialised (low alpha diversity) individuals[33]. It has been further proposed that species with wider niches should exhibit larger levels of individual specialisation coupled with greater among-individual variation in resource use, the so-called niche variation hypothesis[34,35]. Our cross-sectional study design did not enable us to ascertain the dietary specialisation level of each analysed individual, but provided some insights into the structure of dietary diversity. We did not observe a negative linear relationship between dietary niche breadth and individual niche breadth (Fig. 3a; LM: F = 0.10, df = 1,5, $r^2$ = 0.02, p value = 0.76). However, dietary niche breadth (dRER) did exhibit a strong linear relationship with prey turnover across individual bats (Fig. 3b; LM: F = 22.94, df = 1,5, $r^2$ = 0.82, p value = 0.004), showing that dietary niche breadth is shaped by inter-individual variation of diet composition, as the niche variation hypothesis predicts. Prey turnover also exhibited a borderline positive linear trend with distribution homogeneity (Fig. 3c; LM: F = 5.09, df = 1,5, $r^2$ = 0.50, p value = 0.07). This observation suggests that species that exhibit more homogeneous distributions across their ranges exhibit broader dietary niche breadths, probably because they are

able to exploit different types of dietary resources in different environments.

**Dietary niche breadth is associated with hunting plasticity.** Lastly, we analysed the relation of other behavioural traits (hunting plasticity, habitat use diversity and roosting plasticity) with dietary breadth, range size and spatial homogeneity, to explore the links of other niche axes with dietary and spatial features. We found no significant correlation between any of the analysed traits and spatial features, both range size and distribution homogeneity (Supplementary Fig. 9). However, we found that hunting plasticity is positively related with the dietary niche breadth of predator species (Fig. 3d; LMM: t = 3.135, df = 5, $r^2_{marginal}$ = 0.56, $r^2_{conditional}$ = 0.96, p value = 0.026). This suggests that the ability to use a more diverse range of hunting strategies, such as capturing prey from the ground, foliage or water surface, in addition to hunting flying prey[36], also contributes to an obvious broadening of the functional spectrum of captured prey. This increased flexibility could render animals less sensitive to environmental heterogeneity, and thus also contribute to the homogenisation of their spatial distributions.

## Discussion
We combined high-resolution broad-scale dietary (DNA metabarcoding) and spatial (species distribution modelling)

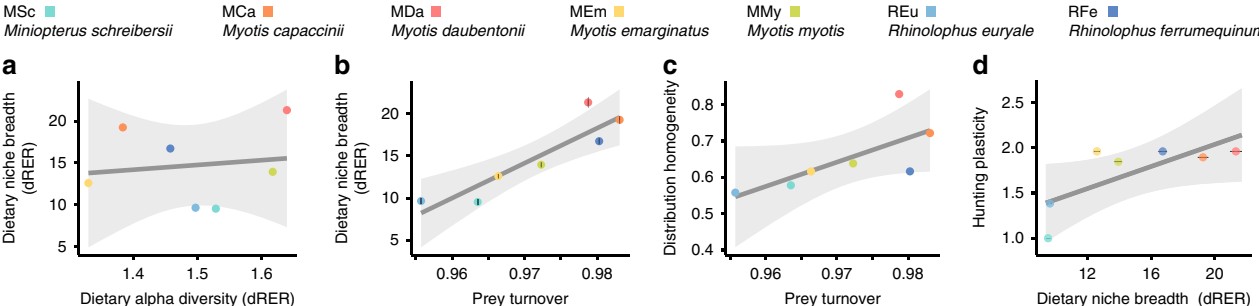

**Fig. 3 Relationships of dietary, spatial and behavioural traits of species. a** Dietary alpha diversity of species shows no linear correlation with dietary niche breadth. **b** Dietary niche breadth exhibits a significant linear correlation with prey turnover. **c** Distribution homogeneity shows a borderline linear trend with prey turnover. **d** Dietary niche breadth exhibits a significant linear correlation with hunting plasticity.

characterisation to relate diet with broad-scale spatial patterns. While dietary niche breadth has been previously shown to predict geographic range sizes in certain taxa[3], dietary niche breadth has not previously been associated with other broad-scale spatial features of species. The ecological niches of bats are complex arrays of different resources, and spatial features are most probably also affected by other niche axes[9,10]. However, the potential noise such confounding factors could have introduced was not strong enough to mask the correlations our data unveiled between dietary breadth and homogeneity of spatial distributions. In addition, after analysing three other Eltonian niche features of bats, only dietary breadth showed significant correlations with spatial features. It is remarkable that a single snapshot of the diet of individual bats was enough to recover a clear link between dietary and spatial features, as niche patterns are not always coupled at individual and population levels[37]. Our results therefore depict diet as one of the most relevant ecological features associated with broad-scale spatial distributions of European bats.

This observation challenges the traditional view on the relevance of Eltonian features at different scales. The ecological significance of the functional traits of species changes across geographic scales, and it is considered that fine-scale features such as resource use (e.g. diet) are "averaged out", and instead, non-interactive variables (e.g. climatic tolerance) determine the broad aspects of distributions of species[38]. However, it has been recently shown that, although strongest at fine grains, interactive variables can remain important even at larger spatial scales[39]. For instance, dietary features have been shown to contribute to shaping broad-scale species distributions in parrots[40]. Our results also show that dietary features are linked to broad-scale spatial features of bats. Our data associate dietary diversification with homogeneity of spatial distribution, through an increased prey turnover across individuals. We also show that dietary breadth is correlated with an increased capacity to employ different hunting strategies. These two observations suggest that the pattern is the result of the combination of the capacity to exploit different prey resources and being exposed to different available prey. The ability to modulate the hunting strategy enables bats exploiting structurally distinct environments, thus eliciting a homogeneous dispersion over the landscape.

It is noteworthy that all the patterns we found in this study were recovered using dietary and spatial distribution metrics that are largely overlooked in the literature. Our dietary breadth metrics achieved explanatory power only when all three components of diversity were considered. No diversity component alone was correlated with spatial features or other axes of the ecological niche of bats, which highlights the importance of accounting for relative evenness and regularity of prey when measuring dietary niches[15,27]. Similarly, the spatial feature that dominates the literature, namely range size, showed no link with

dietary features, while a far less studied feature such as distribution homogeneity did. Our results therefore suggest that exploring ecological patterns beyond simple traditional metrics might unveil associations that otherwise would remain hidden.

Further work will be necessary to ascertain whether the observed patterns are limited to the bat assemblage studied, or can be extended to other taxa and geographic regions. Given the uneven impact climate change is having on species with distinct ecological niches—comparatively favouring generalists over specialists[41,42]—understanding the complex relationships between ecological traits and spatial distribution patterns is of paramount importance for predicting impacts over species with different ecological features. In that regard, integrative approaches that leverage tools and knowledge developed in different fields of biological sciences, such as the one showcased in this study, will be critical for unravelling relevant ecological patterns in the intersection between Eltonian and Grinnellian niches.

## Methods

**Sample collection and dietary data generation**. We collected droppings from 402 individual bats captured in 40 locations distributed across Europe (Supplementary Table 1), in June–October of 2015–2017. The droppings belonged to seven species: *Miniopterus schreibersii* (MSc), *Myotis capaccinii* (MCa), *Myotis daubentonii* (MDa), *Myotis emarginatus* (MEm), *Myotis myotis* (MMy), *Rhinolophus euryale* (REu) and *Rhinolophus ferrumequinum* (RFe). Using a randomised setup, DNA was extracted from all individual samples using the PowerSoil® DNA Isolation Kit (MoBio, CA, USA) principally following the manufacturer's protocol (2016 version), but with some modifications (see Supplementary Information). Extracts were amplified in three replicates using two primer pairs, referred to as Zeale (ZBJ-ArtF1c: AGATATTGGAACWTTATATTTTATTTTTGG; ZBJ-ArtR2c: WACTAATCAATTWCCAAATCCTCC)[43] and Epp (Coleop_16Sc: TGCAAA GGTAGCATAATMATTAG; Coleop_16Sd: TCCATAGGGTCTTCTCGTC)[44], after determining optimal PCR conditions (shown in Supplementary Tables 2 and 3) through quantitative PCR (qPCR) screening. Both primers were 5′ nucleotide tagged to yield a set of unique forward and 60 unique reverse primers. Tags were 7 nucleotides in length and there were 2–3 nucleotide mismatches between tags. Each PCR amplification was carried out with matching tags (e.g. F1-R1, F2-R2, etc.) to ensure tag jumps would not result in false assignments of sequences to samples. The three PCR replicates from each sample were carried out with different tag combinations to minimise the possible effect of tag bias. Each PCR round contained 96 samples, including 90 bat dropping samples, four extraction blanks and two PCR blanks. All PCR mixes were set up in a dedicated pre-PCR laboratory to minimise the risk of contamination. Amplicons were bead-purified, pooled and built into libraries using a single-tube library preparation method[45] modified for amplicon samples. Purified libraries were analysed in an Agilent Bioanalyzer and combined into sequencing pools using equimolar ratios. Library pools were spiked with 15% PhiX before sequencing them in multiple Illumina MiSeq flow cells using 250PE chemistry and aiming 35,000 reads per PCR replicate per sample.

**Dietary metabarcoding bioinformatics procedures**. The paired end reads in each sequencing library were merged and quality filtered using AdapterRemoval 2.1.7[46] and the reads within each library were sorted according to primer and tag sequences using DAMe[47]. To ensure maximum DNA sequence reliability, only high-quality sequences that appeared in at least two of the three PCR replicates were retained, and sequences identical to those detected in the extraction and library blanks of the corresponding processing batch of each sample were removed. Appropriate sampling depth per sample was ensured by discarding samples with

insufficient sequencing depth as assessed by rarefaction curves and curvature indexes. DNA sequences were clustered into operational taxonomic units (OTUs) based on 98% identity following Alberdi et al.[48]. Samples with less than 5000 sequences were removed, and OTUs with a representation below 0.02% in each sample were removed for their probability of being false positives derived from PCR and sequencing errors. Taxonomy was assigned by aligning the OTU representative sequences to the Genbank nt[49]–and in the case of Zeale also BOLD[50]–databases. Bayesian OTU phylogenetic trees were generated using BEAST2 [51] after aligning the OTU representative sequences using CLUSTAL Omega[52]. All the analyses were performed with a minimal Markov chain Monte Carlo (MCMC) chain length of $10^8$ iterations, sampling trees every 1000. Each Bayesian run was repeated, and convergence of the MCMC chains and sample size was checked using TRACER 1.6.0. To account for the phylogenetic uncertainty of the reconstructed trees, 50 trees were randomly selected from the last 5% (5000) of the trees sampled across the MCMC, as detailed by Alberdi et al.[15]. Full details of the bioinformatics methodologies are reported in the Supplementary Information and Supplementary Code 1.

**Dietary diversity analyses.** Diversity analyses were carried out using the R package hilldiv[53] based on abundance-based Hill numbers[54,55]. The Hill numbers framework enables (i) the relative weight given to abundant and rare OTUs to be modulated through a single parameter, namely the order of diversity $q$[54], and (ii) the similarity level across OTUs to be overlooked or accounted for when computing diversity. Although functional diversities can be computed using Hill numbers[56], given the infeasibility of gathering ecological trait information of thousands of prey items, OTU phylogenies were employed as proxies of ecological resemblance across OTU[27]. Hence, dR (richness) was computed as the neutral Hill number of order of diversity $q = 0$; dRE (richness + evenness) was computed as the neutral Hill number of order of diversity $q = 1$–i.e. Shannon diversity–and dRER (richness + evenness + regularity) was computed as the phylogenetic Hill number of order of diversity $q = 1$. Phylogenetic Hill numbers were computed based on Bayesian phylogenies generated from metabarcoding DNA sequences, and the analyses accounted for the phylogenetic uncertainty of generated trees, as detailed in Supplementary Information. Prey turnover was measured by means of Jaccard-type turnover using hilldiv, based on the beta diversity value derived from the multiplicative hierarchical partitioning of the species' dietary diversity into alpha (individuals) and beta (across individuals) components. The Jaccard-type turnover quantifies the normalised prey turnover rate (across individuals) with respect to the whole system (species)[57].

**Spatial distribution analyses.** *Recognised* range sizes were calculated from distribution maps retrieved (2019/09/17) from the IUCN Red List of Threatened Species[32]. Ensemble SDMs were generated for the seven studied bat species using the R package biomod2[58], including four models (MaxEnt 3.4, Generalized Boosting Model, Random Forests, Flexible Discriminant Analysis). Models included between 113 and 591 occurrence records (Supplementary Table 5). Species occurrence records were gathered from the online databases GBIF (www.gbif.org) and EUROBATS (https://www.eurobats.org/), from 33 journal publications (Supplementary Table 7), and unpublished records held by co-authors. To reduce spatial bias and spatial auto-correlation, we first removed low-quality records in terms of spatial resolution and taxonomic identification, and then we used the ArcGIS toolbox "SDMtools"[59] to thin spatially clustered records. We initially considered 36 environmental variables (16 climatic, six geographic, 13 habitat and three human disturbance) to include in the models. We tested for correlation among variables, and selected among highly correlated ones ($|r| > 0.75$) the more ecologically relevant or the variable with the stronger effect on model performance on its own for each bat species. Finally, we discarded variables that did not contribute to model performance. The final set of variables used in the models is shown in Supplementary Tables 5–6. Models were run with 10,000 random background points and 1000 maximum iterations. To assess model performance, we used tenfold cross-validations replicates, with 75% of records retained for training and 25% for model testing. We used area under the curve (AUC) of the receiver operator characteristics and True Skill Statistic (TSS) to evaluate the models (Supplementary Table 5). The ten cross-validated replicates were combined to obtain a final predicted environmental suitability map for each of the four modelling methods. Ensemble models were obtained by using AUC values to proportionally weight each method according to its predictive power, excluding models with AUC < 0.75. Spatial metrics were computed using the breadth function implemented in the R package ENMhill[60]. This function enables computing both absolute and relative spatial breadths from raster data. Potential range size was computed as the number of raster cells above the minimum suitability value estimated for each species. Distribution homogeneity was measured by means of the Hill numbers' evenness factor[61], i.e. through dividing the Hill number of $q = 1$ by the Hill number of $q = 0$. Distribution homogeneity takes the unit when the suitability values of all cells is identical and approaches zero as the suitability distribution of cells becomes more uneven.

**Statistical analyses and tests.** Predator species' hunting strategy, habitat-use and roosting data were gathered from 45 articles available in the literature

(Supplementary Tables 8–10). Breadths of those niche axes were computed by means of Shannon diversity (Hill number of $q = 1$) using the R package hilldiv. Relationships between computed metrics were analysed through regression analyses using linear models (LM) and linear mixed-effect models (LMM). LMs as implemented in the R package stats were used when analysing species-level data with a single value per species. In contrast, LMMs were used when species-level data contained multiple pseudoreplicated measurements, with dietary niche breadth as fixed effect and species as random effect. Such an approach was used when considering dRER values, as dietary breadth values were characterised using 50 different values to account for uncertainty of the prey phylogenetic trees (details in Supplementary Information). The function *lmer* as implemented in the R package lme4[62] was used to compute marginal and conditional $R^2$ values of LMMs. The former describes the proportion of variance explained by the fixed factors, while the latter describes the proportion of variance explained by both the fixed and random factors. $p$ values were fitted and calculated for all effects in the mixed models using the function *mixed* as implemented in the package afex[63]. For all statistical tests, significance threshold was set at $p = 0.05$. All statistical analyses were performed in R[64] after averaging the results yielded by both primers unless otherwise stated. Raw files are stored in Zenodo repository[65].

**Reporting summary.** Further information on research design is available in the Nature Research Reporting Summary linked to this article.

## Data availability

The datasets generated during and/or analysed during the current study are available in Zenodo with the following digital object identifier: https://doi.org/10.5281/zenodo.3610756 (ref. [65]). The source data underlying Figs. 1b–d, 2h–i, and 3a–d are provided as a Source Data file.

## Code availability

The bash, python and R scripts used for analysing the data during the current study are available in the Supplementary Files as Supplementary Code 1 (DNA metabarcoding), Supplementary Code 2 (Species Distribution Modelling) and Supplementary Code 3 (statistical analyses).

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

## Acknowledgements

A.A. was supported by Lundbeckfonden (R250-2017-1351) and the Danish Council for Independent Research (DFF 5051-00033). O.R. was supported by an NERC Independent Research Fellowship (NE/M018660/1), and O.A. was supported by the Carlsberg Foundation's Postdoctoral Fellowship (CF15-0619). M.T.P.G. acknowledges ERC Consolidator Grant (681396-Extinction Genomics). JA and IG were supported by the Spanish and Basque Government grants (CGL2012-38610, CGL2015-69069-P, IT754-13, IT1163-19). We are grateful to Fiona Mathews, Daniel Whitby, Roger Ransome, Matt Cook, Carles Flaquer and Martina Spada for providing samples; and Aitor Arrizabalaga, Lide Jimenez, Vilalii Hukov, Olena Holovchenko, Vanessa Mata and Branka Pejić for assistance in the field work. This article is based upon work from COST Action "CLIMBATS—Climate change and bats: from science to conservation", supported by COST (European Cooperation in Science and Technology).

## Author contributions

A.A., O.A. and M.T.P.G. designed the study. A.A., O.R., O.A., R.N.-F., J.A., I.B., I.G., C.I., E.I., H.R., D.R., A.V., V. Zhelyazkova and V. Zrncic participated in the sample and data collection. A.A. and O.A. performed the laboratory procedures. O.R. and R.N.-F. carried out the species distribution modelling. O.A. performed the ecological trait analyses. A.A. performed the DNA metabarcoding and statistical analyses. A.A. wrote the manuscript. All authors contributed to and approved the final version.

## Competing interests

The authors declare no competing interests.
