## [Peer Review File · Nature Communications]

Reviewers' comments:

Reviewer #1 (Remarks to the Author):

This paper addresses an interesting issue. I am not a modeller and others need to comment on the validity of some of the assumptions made but to me it looks convincing. The conclusion seems to be that there is a significant link between hunting strategy, trophic niche breadth and spatial niche breadth. It suggests there that behavioural plasticity leads to a richer diet and that this expands niche space. For many species (vertebrate and invertebrate) a broad diet brings nutritional benefits and a larger niche space allowing a species to survive in a broad range of habitats. There are of course benefits that can arise from specialist diets too (evolution of more efficient hunting strategies for particular prey species etc.) but this is not considered here.

I have a few criticisms to make, mostly minor. A niche is not simply about diet and hunting strategy. For example the availability of suitable summer and winter roosts can affect distributions. Prey availability is a factor affecting diet richness and diversity (you can only eat what is on offer) and there may be elements of 'prey choice' involved (including factors such as prey escape strategies). However, such 'noise' has not masked the relationships detected in this paper but could be mentioned. Other sources of noise include the fact that the samples were snapshots, taken at very different times of year.

The subtitle 'The trophic niche of European bats is dominated by Lepidoptera and Diptera' needs to be qualified. It is well known that the Zeale primers have a strong bias towards amplifying these taxa but there are many other insects that they do not amplify. This needs reviewing. The assumption has been that as Lepidoptera and Diptera are known to be major prey of insectivorous bats in Europe and elsewhere that this does not matter, and perhaps it doesn't for this modelling exercise. The Epp primers may have similar biases that need checking – hopefully using the two sets have given you the necessary coverage, but this needs to be confirmed. Using two pairs of general invertebrate primers is certainly a plus.

So, overall the work is technically novel in the way it demonstrates that behaviourally adaptable predators eat a wider range of prey species and occupy broader niche space. I agree that using high throughput sequencing allowed this relationship to be studied in far more detail than has been possible before. This is not unexpected, but its good to see this demonstrated so thoroughly.

William O. C. Symondson

Reviewer #2 (Remarks to the Author):

In the manuscript "Diet diversification shapes broad-scale distribution patterns in European bats" (NCOMMS-19-21219), Alberdi et al. investigated whether European bat species' trophic niche properties can predict their respective spatial niche breadth. This is a promising idea that scores high

in originality. Alberdi and colleagues' perspective of dietary niches as potential drivers of large-scale distribution patterns has the potential to bring novel insights on enduring questions in spatial ecology and niche theory. The dataset used is compelling, both in terms of accuracy and spatial extent. I also appreciate a lot the amount of work that the authors took to organize a comprehensive supplementary material & code files to ensure full reproducibility of their results. Data visualization, both in the main text and supplements, is great as well. That being said, I am not entirely sure if the potential of the manuscript is fully explored in the current version. I feel that part of my concerns stems from the too concise format of the manuscript. I strongly suggest that the authors revisit and expand their main text to improve the motivation and framing of their hypotheses/predictions, as well as the presentation and discussion of their findings. Below I list some general and specific considerations, which I hope to be useful to the authors in revising their manuscript.

General considerations

I understand that the authors are space-limited (although Nature Communications accepts contributions up to 5,000 words), but without a clear presentation of the hypotheses (i.e., why and how species' trophic niches are expected to determine large-scale distribution) and their respective biological context, it becomes difficult to contemplate how the results may generalize to other biological systems. I feel that the underlying mechanisms generating a potential link between dietary features and spatial distribution patterns are just superficially presented (lines 61-62). For example, it is not clear how diet richness, evenness, and regularity would impact "the capacity of animals to thrive in a wider range of environmental conditions". Actually, this hypothesis is presented as a derivation from a community-level perspective, i.e. that community-level richness, evenness, and regularity may improve performance (supported by only one reference – Zhang et al. 2012 -, a meta-analysis about forest productivity – line 9). I am afraid this transition between community-level properties to species-level trophic niches is not a trivial one and would require more conceptual and general contextualization.

Moreover, given the concise style of the manuscript, I feel that methodological aspects (i.e., DNA sequencing-based tools) should be avoided in the introduction. In fact, the proportion of the introduction dedicated to explaining the advantages of DNA metabarcoding is relatively high (compared, for example to the theoretical reasons to expect that niche width is associated with geographic distribution). I fully agree that DNA metabarcoding is a fascinating approach and a rich source of robust dietary data, however I think that the introduction should focus on more general aspects how and why diet diversity could shape distribution patterns.

I am also concerned as well about the attempt to establish 'causal directionality' between trophic niche breadth and spatial distribution. I understand the underlying logic supporting the post-hoc tests (i.e., [i] whether diet expansion is driven by beta spatial diversity, and [ii] correlation between richness per se [dR] and spatial niche), however I am not fully convinced that these two additional correlational approaches are indeed enough to "rule out the possibility that dietary breadths are passively broadened as a result of spatial niche expansion" (lines 140-141). The relatively low number of replicates ($n = 7$; number of bat species) does not allow the use of structural equation modeling approaches or more specific causal modeling techniques. Therefore, I strongly recommend

the authors to tone down their statements on assessing the causal directionality of the observed relationships.

The meticulous and spatially wide the characterization of dietary niches in multiple species presented by the authors is very scarce in the literature and certainly it is one of the strengths of the manuscript. On the basis of the presented data, however, I am concerned about one point. Because (i) faecal samples represent a snapshot of the diet and (ii) the average number of individuals/species/site is relatively low (i.e., 355 individual bats, from 7 species, in 40 sites), I wonder if the number of samples (individuals) is enough to characterize well the diet of bat species in each sampling location. A robust description of species' niches at local sites is key to test the major ideas of the manuscript, but it is unclear in the current version if the used data set indeed allows it. Depending on how the hypotheses are framed, I do not think this is a severe problem (and truly believe this is an amazing data set), but I feel the issue mentioned above requires some clarification.

Specific comments

Lines 50-51. The link between trophic niches and large-scale distribution patterns is not obvious, so the authors have to explain how species' diet may affect their distribution. As I explain in my general comments, this is a crucial piece of information that it is missing in the manuscript. Moreover, beyond just saying that the link diet-distribution patterns is still inconclusive, it is important to emphasize why it is inconclusive.

Lines 51-53. This sentence is a bit vague and perhaps a weak way to motivate the problem. Actually, is it possible to quantify all the components of dietary diversity using one or a few diversity metrics?

Lines 57. Not necessarily across much larger sample sizes – stomach contents, stable isotopes, direct observations also allow obtaining sizable sample sizes in many previous dietary studies.

Line 60. This definition of 'richness' in the context of dietary diversity is wrong. Please clarify that it refers to 'how many prey *types*/*species* are consumed'.

Line 61. Please indicate what 'similarity' refers here (e.g., traits, functional, phylogenetic?).

Lines 61-65. As I discuss in my general comments, these two sentences are vital to the narrative but, as it stands, they are superficial. This absence of strong mechanisms that could underlie the effects dietary niches on spatial niches is a structural problem that permeates the whole manuscript. I strongly recommend the authors to reconsider how they motivate their hypothesis and predictions.

Line 69. It would be interesting to briefly mention here the dietary habitats of European bats (e.g. consume a wide diversity of arthropods).

Lines 69-70. Because the idea that behavioural traits may tune the effects of diet on distribution has not been introduced so far, thus it may be hard to readers follow why the authors mention 'behavioural traits' here.

Line 77. I suggest mentioning what the operational metric used to quantify spatial niche breadth is. Is it the potential area of distribution?

Line 92. Please clarify what “species-rank” means – is it prey relative abundances? The legend of figure 1d is also a bit confusing.

Lines 93-95. This combination of sentences is complicating: “were different on the components of diversity considered”, “similar contrasting results”, and “different ecological forces” – I suggest the authors to be more specific here and emphasize the biological meaning of these results.

Lines 98-99. Again, could the authors explain these results in a biological context?

Lines 99-101. But possibility (i) is rejected below in the manuscript (Figure 2c).

Line 119. Please present a quantitative metric revealing the magnitude of variation across species in their ‘spatial projections’ (e.g. range of environmental suitability area?).

Line 120. What are the two spatial niche breadth metrics computed? Figure S4 seems to show two different trophic niche metrics (Zeale and Epp datasets).

Line 123. Because this is a species-level analysis (e.g. spatial niche is a property of species) I do not follow why this correlation has d.f. = 348.

Line 129; 135. “broadened passively” is a bit confusing, please reframe it.

Lines 130-313. “species’ dietary niches”

Lines 130-132. Avoid the distracting repetition of “dietary” – perhaps “trophic” is a good option.

Line 131. I would say that the concept of beta diet diversity is more related to spatial variation (i.e. differences in trophic niche across populations) and not across individuals. When the authors refer to “differences across individuals within species” some readers may interpret it as intrapopulation diet variation (sensu Dan Bolnick studies). Please clarify.

Lines 133-134. It needs to be explained which procedures were made before affirming that the “correlation... remained significant”. It is unclear how the effects of alpha vs. beta diet diversity were considered.

Lines 134. Is a bit confusing that alpha dietary diversity deals with individual diversity. It seems to be true only if one single individual from each species was sampled at each location, which would not be ideal. Please clarify.

Line 143-144. Despite the lack of significant statistical correlation, I wonder how distinct, in conceptual terms, ‘habitat use diversity’ is from ‘spatial niche breadth’.

Line 139-141; 150-151. Given the correlational nature of the analyses, I recommend toning down these statements.

Line 170-171. The distinction between ‘ecological niche breadth’ and ‘trophic niche breadth’ is not a trivial one, I recommend the authors to reframe this sentence and take particular attention when mentioning different facets of the controversial concept of niche.

Lines 172-178. I think these arguments are too speculative given the findings of the study. I suggest the authors to carefully revisit their discussion in light of their findings and tone down it.

I hope the authors find my comments useful in revising their manuscript.

Reviewer #3 (Remarks to the Author):

This is an interesting account of how dietary diversity and spatial distribution are related in European bats. The manuscript is pretty well written, though unclear in some places. In the end, I am left with the impression that what we have learned is fairly narrow. Dietary diversity, in bats, is related to distribution size, in Europe. The discussion should more address the broader concepts used to build a case for the study that are in the introduction.

Specific comments:

1) I think that the correlation analyses are not the way to analyze these data. This is especially true for figures 2a, 2c and 2d. These are all species level analyses yet you have > 300 df when conducting the tests. You are pseudoreplicating and this is inappropriate. I would suggest a mixed model where you have individuals (random effect) nested within species (fixed effect). This will also allow you to assess the within and among species contributions to diversity.

2) How do you determine abundance of the insects that is used to determine hill numbers? Are you using amount of DNA from the barcoding? If so, doesn't size of the insect eaten by the bat affect the amount of DNA in the sample? I don't think you can obtain an unbiased estimate of insect abundance from the genetic work. If so, please justify this in the manuscript.

Lines 63 to 64. This doesn't make sense. If you break this sentence down you hypothesize that these metrics impact the capacity of animals to thrive. These metrics don't affect the bats at all!

Line 68—I don't think there is such a thing as “the European bat community”. Europe likely represents many bat communities. I would use the more generic term of “assemblage”.

Line 235—shannons diversity of ecological traits—please explain or provide a citation.

Reviewer #1 (Remarks to the Author):

This paper addresses an interesting issue. I am not a modeller and others need to comment on the validity of some of the assumptions made but to me it looks convincing. The conclusion seems to be that there is a significant link between hunting strategy, trophic niche breadth and spatial niche breadth. It suggests there that behavioural plasticity leads to a richer diet and that this expands niche space. For many species (vertebrate and invertebrate) a broad diet brings nutritional benefits and a larger niche space allowing a species to survive in a broad range of habitats. There are of course benefits that can arise from specialist diets too (evolution of more efficient hunting strategies for particular prey species etc.) but this is not considered here.

> Following the comments and suggestions of the reviewers, we have detailed the theoretical framework of the study and re-framed the hypothesis under analysis, which now places the discussion between specialist and generalist species in the centre of the article (Lines 55, 70-72, 211-215).

> We better defined the spatial metrics analysed, because the comments of the three reviewers clearly showed that the spatial analyses required further clarification. Specifically, the ‘spatial niche breadth’ metric shown in the old version, has been termed as ‘spatial homogeneity’, because this is what Levins’ indices used in the previous version actually measure. In addition, we have also contrasted our dietary results with range sizes, both recognised (based on IUCN maps) and potential (based on spatial models). We hope the relation between dietary and spatial features is more clear now.

I have a few criticisms to make, mostly minor. A niche is not simply about diet and hunting strategy. For example the availability of suitable summer and winter roosts can affect distributions. Prey availability is a factor affecting diet richness and diversity (you can only eat what is on offer) and there may be elements of ‘prey choice’ involved (including factors such as prey escape strategies). However, such ‘noise’ has not masked the relationships detected in this paper but could be mentioned. Other sources of noise include the fact that the samples were snapshots, taken at very different times of year.

> We have included some texts about this issue in the discussion (Lines 251-256).

“The ecological niches of bats are complex arrays of different resources, and spatial features are most probably also affected by other niche axes. However, the potential noise such confounding factors could have introduced was not strong enough to mask the correlations our data unveiled between dietary breadth and homogeneity of spatial distributions...”

The subtitle ‘The trophic niche of European bats is dominated by Lepidoptera and Diptera’ needs to be qualified. It is well known that the Zeale primers have a strong bias towards amplifying these taxa but there are many other insects that they do not amplify. This needs reviewing. The assumption has been that as Lepidoptera and Diptera are known to be major prey of insectivorous bats in Europe and elsewhere that this does not matter, and perhaps it doesn’t for this modelling exercise. The Epp primers may have similar biases that need checking – hopefully using the two sets have given you the necessary coverage, but this needs to be confirmed. Using two pairs of general invertebrate primers is certainly a plus.

> We have now added a supplementary figure (Figure S5) to evaluate primer amplification biases and have briefly discussed the associated methodological issues (Lines 127-132) without losing the general ecological script.

“The reliability of the results was further confirmed through *in silico* taxonomic amplification bias analyses, which showed that Epp primers do not exhibit the bias towards Lepidoptera and Diptera (Fig. S5) that is known to affect the Zeale primers.”

So, overall the work is technically novel in the way it demonstrates that behaviourally adaptable predators eat a wider range of prey species and occupy broader niche space. I agree that using high throughput sequencing allowed this relationship to be studied in far more detail than has been possible before. This is not unexpected, but it's good to see this demonstrated so thoroughly.

William O. C. Symondson

> Thank you very much, Prof. Symondson.

Reviewer #2 (Remarks to the Author):

In the manuscript “Diet diversification shapes broad-scale distribution patterns in European bats” (NCOMMS-19-21219), Alberdi et al. investigated whether European bat species' trophic niche properties can predict their respective spatial niche breadth. This is a promising idea that scores high in originality. Alberdi and colleagues' perspective of dietary niches as potential drivers of large-scale distribution patterns has the potential to bring novel insights on enduring questions in spatial ecology and niche theory. The dataset used is compelling, both in terms of accuracy and spatial extent. I also appreciate a lot the amount of work that the authors took to organize a comprehensive supplementary material & code files to ensure full reproducibility of their results. Data visualization, both in the main text and supplements, is great as well. That being said, I am not entirely sure if the potential of the manuscript is fully explored in the current version. I feel that part of my concerns stems from the too concise format of the manuscript. I strongly suggest that the authors revisit and expand their main text to improve the motivation and framing of their hypotheses/predictions, as well as the presentation and discussion of their findings. Below I list some general and specific considerations, which I hope to be useful to the authors in revising their manuscript.

> Thank you. We have expanded the main text as suggested. See response to comments below.

General considerations

I understand that the authors are space-limited (although Nature Communications accepts contributions up to 5,000 words), but without a clear presentation of the hypotheses (i.e., why and how species' trophic niches are expected to determine large-scale distribution) and their respective biological context, it becomes difficult to contemplate how the results may generalize to other biological systems. I feel that the underlying mechanisms generating a potential link between dietary features and spatial distribution patterns are just superficially presented (lines 61-62). For example, it is not clear how diet richness,

evenness, and regularity would impact “the capacity of animals to thrive in a wider range of environmental conditions”. Actually, this hypothesis is presented as a derivation from a community-level perspective, i.e. that community-level richness, evenness, and regularity may improve performance (supported by only one reference – Zhang et al. 2012 -, a meta-analysis about forest productivity – line 9). I am afraid this transition between community-level properties to species-level trophic niches is not a trivial one and would require more conceptual and general contextualization.

> We agree with the reviewer that some necessary details were omitted to create a short and easy-to-read manuscript. After reading the three reviewers’ comments, it is clear that there was an overall lack of detail in the explanation of the “spatial niche breadth”. We have in consequence changed the terminology to spatial homogeneity (as mentioned before, this is exactly what the original analyses measured), and we have also included a contrast between distribution range size and spatial homogeneity (Lines 77-83, 116-120).

“Besides range size, other spatial features might also be related to dietary niche breadth. The homogeneity of the spatial distribution, i.e. how evenly animals are distributed within their geographic range, is one of them. (...)”

“The contrast between different components of dietary diversity and spatial patterns enabled us to unveil relationships between dietary niche breadth and broad-scale spatial features, and test whether the dietary-niche-breadth::distribution-homogeneity hypothesis stands in the studied European bat assemblage.”

> All in all, we have now expanded the rationale of our main hypothesis (Lines 68-95) in the introduction and discuss it in further depth in the discussion (Lines 262-276).

“While the dietary-niche-breadth::range-size hypothesis has been thoroughly studied, dietary-niche-breadth::distribution-homogeneity hypothesis has received far less attention.”

“Our data associate dietary diversification with homogeneity of spatial distribution, through an increased prey turnover across individuals. We also show that dietary breadth is correlated with an increased capacity to employ different hunting strategies. These two observations suggest that the pattern is the result of the combination of the capacity to exploit different prey resources and being exposed to different available prey. The ability to modulate the hunting strategy enables bats exploiting structurally distinct environments, thus yielding a homogeneous dispersion over the landscape.”

> We have also included a short discussion about transitioning from community-level properties to species-level trophic niches (Lines 262-269).

“Our results also challenge the traditional view on the relevance of Eltonian features at different scales. The ecological significance of the functional traits of species changes across scales, and it

is considered that fine-scale features such as resource use (e.g. diet) are “averaged out”, and instead, non-interactive variables (e.g. climatic tolerance) determine the broad aspects of distributions of species .”

Moreover, given the concise style of the manuscript, I feel that methodological aspects (i.e., DNA sequencing-based tools) should be avoided in the introduction. In fact, the proportion of the introduction dedicated to explaining the advantages of DNA metabarcoding is relatively high (compared, for example to the theoretical reasons to expect that niche width is associated with geographic distribution). I fully agree that DNA metabarcoding is a fascinating approach and a rich source of robust dietary data, however I think that the introduction should focus on more general aspects how and why diet diversity could shape distribution patterns.

> We believe it is necessary to mention the methodological issues in the introduction, because the reason why the pattern we observed has not been so far reported is most probably due to methodological or analytical limitations as we mention in Lines 85-88. Nevertheless, we have shortened the information on methods in the introduction and substantially increased the text devoted to explaining the link between species’ broad-scale dietary diversity and spatial patterns.

“We argue that because traditionally employed metrics are excessively simplified, they might be unable to reveal ecologically meaningful relationships between dietary niche features and broad-scale spatial patterns in vertebrates. However, new analytical tools might provide the required amplitude and resolution to unravel more complex links.”

I am also concerned as well about the attempt to establish ‘causal directionality’ between trophic niche breadth and spatial distribution. I understand the underlying logic supporting the post-hoc tests (i.e., [i] whether diet expansion is driven by beta spatial diversity, and [ii] correlation between richness per se [dR] and spatial niche), however I am not fully convinced that these two additional correlational approaches are indeed enough to “rule out the possibility that dietary breadths are passively broadened as a result of spatial niche expansion” (lines 140-141). The relatively low number of replicates ($n = 7$; number of bat species) does not allow the use of structural equation modeling approaches or more specific causal modeling techniques. Therefore, I strongly recommend the authors to tone down their statements on assessing the causal directionality of the observed relationships.

> We agree with the reviewer and the editor that our data and analyses do not enable to prove causal directionality. Therefore, we have modified the title and toned down our statements.

“DNA metabarcoding and spatial modelling link diet diversification with distribution homogeneity in European bats”

> We have also improved the statistical analyses to account for the nestedness of the data, by implementing linear mixed models as suggested by Reviewer 3 (Lines 201, 352...), and performed power analyses and printed power plots to assess the validity of using structural equation modelling with our data:

> Those analyses estimated that a predator-species number of 10 would provide a power of >95%. This is, in fact, the number of species we initially aimed for, but we decided to discard three species because we were unable to obtain samples from more than 40 individual animals from 8 sites, the minimum required, in our opinion, to appropriately characterise the dietary niche of a species. Although recognising power (73%) of our contrast between dietary niche breadth (dRER) and spatial homogeneity is slightly below ideal, given the complexity of the data and the difficulties to improve sample size due to logistical and conservation issues, we consider effect size is high enough to implement a linear (mixed) model approach.

The meticulous and spatially wide the characterization of dietary niches in multiple species presented by the authors is very scarce in the literature and certainly it is one of the strengths of the manuscript. On the basis of the presented data, however, I am concerned about one point. Because (i) faecal samples represent a snapshot of the diet and (ii) the average number of individuals/species/site is relatively low (i.e., 355 individual bats, from 7 species, in 40 sites), I wonder if the number of samples (individuals) is enough to characterize well the diet of bat species in each sampling location. A robust description of species' niches at local sites is key to test the major ideas of the manuscript, but it is unclear in the current version if the used data set indeed allows it. Depending on how the hypotheses are framed, I do not think this is a severe problem (and truly believe this is an amazing data set), but I feel the issue mentioned above requires some clarification.

> We agree with the reviewer that larger sample sizes for each species within each sampling location would have been better for characterising the local diet of each bat species, and that our dataset only covers a fraction of the dietary complexity of these seven species. It is however important to highlight that all these seven bat species are endangered and highly protected under national and international law, and consequently, it was impossible to capture more than 5 animals per species in some locations, which limited our sample sizes. That is one of the reasons why we opted for using an expansive approach trying

to cover as much spatial and temporal variability as possible, acknowledging the large degree of ecological “noise” of the dataset.

> Consequently, our aim was not to provide a thorough diet inventory of the seven species, but to test a hypothesis that we hoped would have a strong enough signal to surface above this noise, as it was finally shown. Hence, while acknowledging the limitations, as the reviewer mentions in the end, we do not think this limitations disprove the validity of our findings.

Specific comments

Lines 50-51. The link between trophic niches and large-scale distribution patterns is not obvious, so the authors have to explain how species’ diet may affect their distribution. As I explain in my general comments, this is a crucial piece of information that it is missing in the manuscript. Moreover, beyond just saying that the link diet-distribution patterns is still inconclusive, it is important to emphasize why it is inconclusive.

> We agree that the theoretical rationale of the study was not appropriately presented, as a result of a simplification and abbreviation process to fit the most relevant contents in a short manuscript. We have now better defined the hypothesis and expanded the introduction to better explain its rationale. We have also added some text mentioning the likely reason why diet-distribution patterns in vertebrates are still inconclusive (Lines 70-75).

“In addition, vertebrates often exhibit mismatches between different niche components; they can be generalists for dietary resources, but specialists for some other resources, such as roosts. These discrepancies could lead to discordance between given Eltonian features and range size. Finally, range size is also known to be largely dependent on the historic legacy of speciation, Quaternary climatic changes, and the dispersal capacity of species. (...) Besides range size, other spatial features might also be related to dietary niche breadth. The homogeneity of the spatial distribution, i.e. how evenly animals are distributed within their geographic range, is one of them.”

Lines 51-53. This sentence is a bit vague and perhaps a weak way to motivate the problem. Actually, is it possible to quantify all the components of dietary diversity using one or a few diversity metrics?

> This sentence has been removed from the updated manuscript. Regarding the question, it is probably impossible to quantify ALL components of dietary diversity using a few diversity metrics, but it is definitely possible to quantify some of the most relevant ones, as we show in the manuscript. As we mention in the text (Lines 278-287), regularity has been overlooked in most diet-based diversity analyses, and our results show that an important part of information is lost if (dis)similarities across prey are not considered.

“It is noteworthy that all the patterns we found in this study were recovered using dietary and spatial distribution metrics that are largely overlooked in the literature. Our dietary breadth

metrics achieved explanatory power only when all three components of diversity were considered.”

Lines 57. Not necessarily across much larger sample sizes – stomach contents, stable isotopes, direct observations also allow obtaining sizable sample sizes in many previous dietary studies.

> We have rewritten the sentence (Lines 88-91).

“High-throughput DNA sequencing-based (DNA metabarcoding) and species distribution modelling (SDM) tools now enable more nuanced analysis of dietary and spatial patterns. (...)”

Line 60. This definition of ‘richness’ in the context of dietary diversity is wrong. Please clarify that it refers to ‘how many prey *types*/*species* are consumed’.

> We agree that the sentence was not accurate enough, as it was unclear whether it referred to a quantitative or qualitative measurement. We have now specified we refer to “prey types” (Line 92).

Line 61. Please indicate what ‘similarity’ refers here (e.g., traits, functional, phylogenetic?).

> It can be any of those listed by the reviewer. In our case it was phylogenetic because it is impossible to gather ecological traits of >3000 prey belonging to 29 orders, as we mentioned in the methods section. However, ecological-trait based analyses are also possible when the niche breadth is limited to morphologically well-characterised prey, e.g. Arrizabalaga et al. 2019 J Anim Ecol.

Lines 61-65. As I discuss in my general comments, these two sentences are vital to the narrative but, as it stands, they are superficial. This absence of strong mechanisms that could underlie the effects dietary niches on spatial niches is a structural problem that permeates the whole manuscript. I strongly recommend the authors to reconsider how they motivate their hypothesis and predictions.

> As mentioned several times before, the theoretical rationale of the study has been re-framed and the manuscript adapted accordingly.

Line 69. It would be interesting to briefly mention here the dietary habitats of European bats (e.g. consume a wide diversity of arthropods).

> We have modified the text (Lines 101-103).

“The dietary breadth of European insectivorous bats varies considerably, from specialists on certain arthropods —usually moths— to generalists that consume a wide range of taxa.”

Lines 69-70. Because the idea that behavioural traits may tune the effects of diet on distribution has not been introduced so far, thus it may be hard to readers follow why the authors mention ‘behavioural traits’ here.

> This text has been removed from the revised manuscript.

Line 77. I suggest mentioning what the operational metric used to quantify spatial niche breadth is. Is it the potential area of distribution?

> As mentioned before, we have now better defined the analysed spatial metrics. First, we removed the “spatial niche breadth” concept from the manuscript, for considering too vague and potentially misleading. We have changed the name of the original metric to “spatial homogeneity” for considering more accurate, and we have included novel analyses based on range sizes (Lines 112-120).

“we generated species distribution models of the bat species, using high-quality presence records and an ensemble approach that combined different modelling algorithms. Different features of the spatial niche were also characterised based on Hill numbers, so that not only spatial extension was considered, but also the degree of homogeneity of the predicted suitable distributions.”

> In the revised version, we for the first time implemented the Hill numbers into the niche metrics based on distribution models, and modified the text accordingly. Hill numbers can be used to analyse both data sets, and we think it is more elegant and clear to apply the same statistical framework to both data sets. The change is mainly conceptual, as the Levins’ B1 metric we used in the original version was actually a homogeneity factor defined by Hill numbers. Further explanations about the spatial metrics have been added to the methods section (Lines 341-347).

“Spatial metrics were computed using the breadth function implemented in the R package ENMhill. This function enables computing both absolute and relative spatial breadths from raster data. Potential range size was computed as the number of raster cells above the minimum suitability value estimated for each species. Distribution homogeneity was measured by means of the Hill numbers’ evenness factor 50, i.e. through dividing the Hill number of $q=1$ by the Hill number of $q=0$. Distribution homogeneity takes the unit when the suitability values of all cells is identical and approaches zero as the suitability distribution of cells becomes more uneven.”

Line 92. Please clarify what “species-rank” means – is it prey relative abundances? The legend of figure 1d is also a bit confusing.

> It is simply the rank of predator species from the one with the widest dietary niche breadth (generalist) to the one with the narrowest breadth (specialist). We have modified the main text (Line 140), Figure 1 and its footnote (Lines 163-164) to increase clarity.

Lines 93-95. This combination of sentences is complicating: “were different on the components of diversity considered”, “similar contrasting results”, and “different ecological forces” – I suggest the authors to be more specific here and emphasize the biological meaning of these results.

> The sentences have been modified and simplified for the sake of clarity (Lines 141-142).

“were different depending on the components of diversity considered for quantifying dietary niches. Such a dependency on the considered diversity components has also been reported in other systems, which has been attributed to the fact that each diversity component might be driven by different ecological forces”

Lines 98-99. Again, could the authors explain these results in a biological context?

>The biological explanation was (and is) explained in the following lines.

"These differences could be explained by..."

Lines 99-101. But possibility (i) is rejected below in the manuscript (Figure 2c).

> The data rejects the correlation between dR and spatial homogeneity (former “spatial niche breadth”, which does not rule out richness to be driven by other ecological factors. Here we mention one possible feature, which is home range. We preferred not to further develop this issue, to avoid losing the storyline and making the manuscript too complicated.

Line 119. Please present a quantitative metric revealing the magnitude of variation across species in their ‘spatial projections’ (e.g. range of environmental suitability area?).

> In the original version, as it was mentioned in the methods, spatial niche breadth was quantified by means of Levins’ B1 and B2 metrics. In the new version though, we have implemented Hill numbers to measure different features of spatial distributions of species, as explained in the methods section (Lines 341-347).

Line 120. What are the two spatial niche breadth metrics computed? Figure S4 seems to show two different trophic niche metrics (Zeale and Epp datasets).

> We have now modified the analyses and related texts. The dietary niche metrics yielded by Zeale and Epp datasets are slightly different (now in Fig. S8, because the data were different (Zeale targets the COI gene while Epp targets the 16S gene).

Line 123. Because this is a species-level analysis (e.g. spatial niche is a property of species) I do not follow why this correlation has d.f. = 348.

> Following the suggestions of Rev#3, the statistical analyses were modified to linear mixed models. Further explanations can be found below.

Line 129; 135. “broadened passively” is a bit confusing, please reframe it.

> The concept has been removed from the revised manuscript.

Lines 130-313. “species’ dietary niches”

> Corrected.

Lines 130-132. Avoid the distracting repetition of “dietary” – perhaps “trophic” is a good option.

> We avoided using the term “trophic” for considering a concept with a broader meaning than diet, as it might also incorporate hunting and foraging features, and to avoid misunderstandings.

Line 131. I would say that the concept of beta diet diversity is more related to spatial variation (i.e. differences in trophic niche across populations) and not across individuals. When the authors refer to “differences across individuals within species” some readers may interpret it as intrapopulational diet variation (sensu Dan Bolnick studies). Please clarify.

> We used the term beta diversity as a purely mathematical concept here (the result of hierarchical partitioning of diversity). We modified the text to clarify this issue (Lines 208-212).

“We decomposed dietary diversity into alpha (individual bats), beta and gamma (bat species) components to gain insights into the sources of variation across species, because generalist species can be heterogeneous collections (high beta diversity) of specialised (low alpha diversity) individuals . ”

> We have also added further explanations to the methods (Lines 316-320):

“Prey turnover was measured by means of Jaccard-type turnover using *hilldiv*, based on the beta diversity value derived from the multiplicative hierarchical partitioning of the species’ dietary diversity into alpha (individuals) and beta (across individuals) components. The Jaccard-type turnover quantifies the normalized prey turnover rate (across individuals) with respect to the whole system (species).”

Lines 133-134. It needs to be explained which procedures were made before affirming that the “correlation... remained significant”. It is unclear how the effects of alpha vs. beta diet diversity were considered.

> This analysis has been removed from the revised manuscript.

Lines 134. Is a bit confusing that alpha dietary diversity deals with individual diversity. It seems to be true only if one single individual from each species was sampled at each location, which would not be ideal. Please clarify.

> We have now better explained the hierarchical partitioning framework employed in the study (Lines 208-212, 316-320). Although the analysis could be conducted using multiple hierarchical levels, we opted

to avoid it because interindividual prey turnover exhibited a strong linear correlation with dietary breadth, and adding multiple hierarchical levels would excessively complicate the analysis. We think it is logical at first sight to think that animals sampled in a single site are autocorrelated. However, it must be noted that most of these bats have foraging ranges of up to 30-35 km, often show solitary hunting strategies, and commonly switch between roosts. Therefore, the correlating effect of the roosting site is lower than intuitively expected.

> However, if the editor considers such multi-hierarchy approach is necessary, we will gladly include it in the manuscript.

Line 143-144. Despite the lack of significant statistical correlation, I wonder how distinct, in conceptual terms, ‘habitat use diversity’ is from ‘spatial niche breadth’.

> The ‘spatial niche breadth’ concept has been removed from the revised manuscript.

Line 139-141; 150-151. Given the correlational nature of the analyses, I recommend toning down these statements.

> We have removed those sentences from the manuscript.

Line 170-171. The distinction between ‘ecological niche breadth’ and ‘trophic niche breadth’ is not a trivial one, I recommend the authors to reframe this sentence and take particular attention when mentioning different facets of the controversial concept of niche.

> We have modified this sentence to clarify it (Lines 249-251).

“While dietary niche breadth has been previously shown to predict geographic range sizes in certain taxa, dietary niche breadth has not previously been associated with other broad-scale spatial features of species.”

Lines 172-178. I think these arguments are too speculative given the findings of the study. I suggest the authors to carefully revisit their discussion in light of their findings and tone down it.

> These sentences have been removed from the revised version of the manuscript/

I hope the authors find my comments useful in revising their manuscript.

> Yes, indeed. Thank you very much for your thorough and constructive review.

Reviewer #3 (Remarks to the Author):

This is an interesting account of how dietary diversity and spatial distribution are related in European bats. The manuscript is pretty well written, though unclear in some places. In the end, I am left with the

impression that what we have learned is fairly narrow. Dietary diversity, in bats, is related to distribution size, in Europe. The discussion should more address the broader concepts used to build a case for the study that are in the introduction.

> We have expanded the discussion to address the broader ecological concepts as suggested by the reviewer. It must be noted though that dietary diversity is not related to distribution size, but spatial homogeneity. We have clarified this distinction in the new version (Lines 77-83, 112-120 and 167-179, for example).

“Different features of the spatial niche were also characterised based on Hill numbers, so that not only spatial extension was considered, but also the degree of homogeneity of the predicted suitable distributions”

“Similar to dietary niche features, spatial features of species can also be measured through different metrics. To understand the relationship between the different spatial properties, we calculated the most commonly employed feature, namely range size, using IUCN cartography, hereafter referred to as recognised range size. We contrasted it with two other spatial features predicted through species’ distribution modelling (Table S5), namely potential range size and spatial homogeneity of the distribution (Fig. 2A).”

Specific comments:

1) I think that the correlation analyses are not the way to analyze these data. This is especially true for figures 2a, 2c and 2d. These are all species level analyses yet you have > 300 df when conducting the tests. You are pseudoreplicating and this is inappropriate. I would suggest a mixed model where you have individuals (random effect) nested within species (fixed effect). This will also allow you to assess the within and among species contributions to diversity.

> We agree with the reviewer that the statistical approach employed for some analyses was not correct. Species-level analyses that do not include regularity (i.e. dR and dRE) have 5 df, because they are based on weighted gamma dietary diversities of seven species as computed using Hill numbers. However, species-level dRER has 348 df because 50 iterations with different phylogenetic trees from the Bayesian MCMC were used to generate species’ dRER values. We agree that in this latter case a mixed linear model would be more appropriate to account for the pseudoreplication issue mentioned by the reviewer. Individual-based analyses clustered by species also face the same issue. Hence, we have modified those analysis by incorporating mixed linear models as implemented in *lme4* and computing marginal and conditional r-squared values using *piecewiseSEM* (Lines 351-355).

“Relationships between computed metrics were analysed using linear models, although linear mixed models, as implemented in the R packages *lme4* and *afex*, were used when considering dRER, as dietary breadth values were characterised using different values to account for uncertainty of the prey phylogenetic trees (details in Supplementary Information).”

2) How do you determine abundance of the insects that is used to determine hill numbers? Are you using amount of DNA from the barcoding? If so, doesn't size of the insect eaten by the bat affect the amount of DNA in the sample? I don't think you can obtain an unbiased estimate of insect abundance from the genetic work. If so, please justify this in the manuscript.

> We actually implemented both approaches, but only reported the abundance-based results for the sake of clarity and because it enabled obtaining individual-level insights. In the revised version, we have explicitly mentioned that we used both approaches (e.g. lines 111-112, 128-129 and 159), and also included both results (Fig. S6, S7, S8).

“(…) and applying both incidence-based and abundance-based approaches to quantify diet.”

“This pattern was consistent for data generated with both primer sets (Fig. S3), as well as both incidence- and abundance-based approaches employed to quantify dietary profiles (Fig. S4).”

”The incidence-based figure is available as Fig. S4 in the Supplementary Information. ”

> As it can be observed in those figures, despite their different biases (none of the approaches is free of introducing distortion), both approaches yield very similar results, and the overall patterns do not change. This is another example of the robustness of our results.

Lines 63 to 64. This doesn't make sense. If you break this sentence down you hypothesize that these metrics impact the capacity of animals to thrive. These metrics don't affect the bats at all!

> This sentence has been removed from the revised manuscript.

Line 68—I don't think there is such a thing as “the European bat community”. Europe likely represents many bat communities. I would use the more generic term of “assemblage”.

> We have modified it (Lines 97-98).

“In the present study, we contrasted broad-scale dietary and spatial features of a vertebrate system, namely a European bat assemblage”

Line 235—shannons diversity of ecological traits—please explain or provide a citation.

> We have modified the text to make it more comprehensive (Line 350-351).

Reviewers' comments:

Reviewer #2 (Remarks to the Author):

In the revised manuscript 'DNA metabarcoding and spatial modelling link diet diversification with distribution homogeneity in European bats', Alberdi et al. investigated the promising but little-studied link between dietary niches and spatial distribution properties in seven species of Old World bats. I reviewed a previous version of this manuscript, and I am positively impressed by the extensive changes made by the authors in this revised version. I particularly appreciate the amount of work that the authors took to clarify/update the statistical analyzes. Overall, I think the manuscript had improved substantially and could be a good contribution to Nature Communications. I have only very few additional minor comments.

###

Specific comments

Line 39: Thinking on readers that will look only at the abstract, I suggest including a few examples "(e.g., ...)" of which simple metrics are you referring here.

Line 64. Please expand a bit on how "tolerance" is related to Grinnellian niches (e.g. thermal tolerances).

Line 64. Please add citations to support this idea.

Line 83. I am not sure if the nomenclature/notation "dietary-niche-breadth::distribution-homogeneity" is the most appropriate for a broad audience. I feel this notation is distracting and a bit confusing.

Lines 89-90. Change one of the "enable" by "allow" to avoid repetition.

Line 103. Although it may sound a bit repetitive, I would recommend the authors to explicitly add their predictions here (e.g. "We expect that bat species...").

Lines 114-117. "Different" appears three times here; please change one of them by a synonym.

Lines 278-287. I agree that this “methodological” perspective is an important one and has to be present in the discussion. However, I would expect a closing paragraph with more general aspects on how the authors’ findings add to the little-explored intersection between Eltonian & Grinnellian niches.

Figures: Again, the figures are fantastic!

In the revised manuscript 'DNA metabarcoding and spatial modelling link diet diversification with distribution homogeneity in European bats', Alberdi et al. investigated the promising but little-studied link between dietary niches and spatial distribution properties in seven species of Old World bats. I reviewed a previous version of this manuscript, and I am positively impressed by the extensive changes made by the authors in this revised version. I particularly appreciate the amount of work that the authors took to clarify/update the statistical analyzes. Overall, I think the manuscript had improved substantially and could be a good contribution to Nature Communications. I have only very few additional minor comments.

> Thank you very much for the positive feedback.

###

Specific comments

Line 39: Thinking on readers that will look only at the abstract, I suggest including a few examples "(e.g., ...)" of which simple metrics are you referring here.

> Done (Lines 38-39).

"(...) on simple metrics such as low-resolution dietary breadth and range size, which might have (...)"

Line 64. Please expand a bit on how "tolerance" is related to Grinnellian niches (e.g. thermal tolerances).

> Done (Line 64).

"(...) Grinnellian (e.g. climatic tolerance) to Eltonian (...)"

Line 64. Please add citations to support this idea.

> Done (Line 64).

Line 83. I am not sure if the nomenclature/notation "dietary-niche-breadth::distribution-homogeneity" is the most appropriate for a broad audience. I feel this notation is distracting and a bit confusing.

> Modified (Lines 82-83).

"(...) While the hypothesis that links dietary niche breadth with range-size has been thoroughly studied, the hypothesis that relates dietary niche breadth with distribution homogeneity has received far less attention (...)"

Lines 89-90. Change one of the "enable" by "allow" to avoid repetition.

> Modified (Line 91).

Line 103. Although it may sound a bit repetitive, I would recommend the authors to explicitly add their predictions here (e.g. “We expect that bat species...”).

> Added (Lines 104-106).

“(…) Specifically, we hypothesised that the predicted geographic distributions of species would be more homogeneous in bats with broader dietary niches. (…)”

Lines 114-117. “Different” appears three times here; please change one of them by a synonym.

> We modified the structure of the sentence to make it more clear (Lines 117-119).

“(…) Spatial features derived from these models, such as potential range size and degree of homogeneity of the predicted suitable distributions, were also characterised based on Hill numbers applied to spatial data. (…)”

Lines 278-287. I agree that this “methodological” perspective is an important one and has to be present in the discussion. However, I would expect a closing paragraph with more general aspects on how the authors’ findings add to the little-explored intersection between Eltonian & Grinnellian niches.

> We have included a new last paragraph (Lines 292-300).

“Further work will be necessary to ascertain whether the observed patterns are limited to the bat assemblage studied, or can be extended to other taxa and geographic regions. Given the uneven impact climate change is having on species with distinct ecological niches – comparatively favouring generalists over specialists, understanding the complex relationships between ecological traits and spatial distribution patterns is of paramount importance for predicting impacts over species with different ecological features. In that regard, integrative approaches that leverage tools and knowledge developed in different fields of biological sciences, such as the one showcased in this study, will be critical for unravelling relevant ecological patterns in the intersection between Eltonian and Grinnellian niches.”

Figures: Again, the figures are fantastic!

> Thank you!